# How regularization affects the critical points in linear networks

**Amirhossein Taghvaei**[*]
Coordinated Science Laboratory
University of Illinois at Urbana-Champaign
Urbana, IL, 61801
taghvae2@illinois.edu

**Jin W. Kim**
Coordinated Science Laboratory
University of Illinois at Urbana-Champaign
Urbana, IL, 61801
jkim684@illinois.edu

**Prashant G. Mehta**
Coordinated Science Laboratory
University of Illinois at Urbana-Champaign
Urbana, IL, 61801
mehtapg@illinois.edu

## Abstract

This paper is concerned with the problem of representing and learning a linear transformation using a linear neural network. In recent years, there is a growing interest in the study of such networks, in part due to the successes of deep learning. The main question of this body of research (and also of our paper) is related to the existence and optimality properties of the critical points of the mean-squared loss function. An additional primary concern of our paper pertains to the robustness of these critical points in the face of (a small amount of) regularization. An optimal control model is introduced for this purpose and a learning algorithm (backprop with weight decay) derived for the same using the Hamilton's formulation of optimal control. The formulation is used to provide a complete characterization of the critical points in terms of the solutions of a nonlinear matrix-valued equation, referred to as the characteristic equation. Analytical and numerical tools from bifurcation theory are used to compute the critical points via the solutions of the characteristic equation.

## 1 Introduction

This paper is concerned with the problem of representing and learning a linear transformation with a linear neural network. Although a classical problem (Baldi and Hornik [1989, 1995]), there has been a renewed interest in such networks (Saxe et al. [2013], Kawaguchi [2016], Hardt and Ma [2016], Gunasekar et al. [2017]) because of the successes of deep learning. The motivation for studying linear networks is to gain insight into the optimization problem for the more general nonlinear networks. A

---

[*]Financial support from the NSF CMMI grant 1462773 is gratefully acknowledged.

focus of the recent research on these (and also nonlinear) networks has been on the analysis of the critical points of the non-convex loss function (Dauphin et al. [2014], Choromanska et al. [2015a,b], Soudry and Carmon [2016], Bhojanapalli et al. [2016]). This is also the focus of our paper.

**Problem:** The input-output model is assumed to be of the following linear form:

$$Z = RX_0 + \xi \tag{1}$$

where $X_0 \in \mathbb{R}^{d \times 1}$ is the input, $Z \in \mathbb{R}^{d \times 1}$ is the output, and $\xi \in \mathbb{R}^{d \times 1}$ is the noise. The input $X_0$ is modeled as a random variable whose distribution is denoted as $p_0$. Its second moment is denoted as $\Sigma_0 := \mathsf{E}[X_0 X_0^\top]$ and assumed to be finite. The noise $\xi$ is assumed to be independent of $X_0$, with zero mean and finite variance. The linear transformation $R \in M_d(\mathbb{R})$ is assumed to satisfy a property (P1) introduced in Sec. 3 ($M_d(\mathbb{R})$ denotes the set of $d \times d$ matrices). The problem is to learn the weights of a linear neural network from i.i.d. input-output samples $\{(X_0^k, Z^k)\}_{k=1}^K$.

**Solution architecture:** is a continuous-time linear feedforward neural network model:

$$\frac{\mathrm{d}X_t}{\mathrm{d}t} = A_t X_t \tag{2}$$

where $A_t \in M_d(\mathbb{R})$ are the network weights indexed by continuous-time (surrogate for layer) $t \in [0, T]$, and $X_0$ is the initial condition at time $t = 0$ (same as the input data). The parameter $T$ denotes the network depth. The optimization problem is to choose the weights $A_t$ over the time-horizon $[0, T]$ to minimize the mean-squared loss function:

$$\mathsf{E}[|X_T - Z|^2] \tag{3}$$

This problem is referred to as the $[\lambda = 0]$ problem.

Backprop is a stochastic gradient descent algorithm for learning the weights $A_t$. In general, one obtains (asymptotic) convergence of the learning algorithm to a (local) minimum of the optimization problem Lee et al. [2016], Ge et al. [2015]. This has spurred investigation of the critical points of the loss function (3) and the optimality properties (local vs. global minima, saddle points) of these points. For linear multilayer (discrete) neural networks (MNN), strong conclusions have been obtained under rather mild conditions: every local minimum is a global minimum and every critical point that is not a local minimum is a saddle point Kawaguchi [2016], Baldi and Hornik [1989]. For the discrete counterpart of the $[\lambda = 0]$ problem (referred to as the linear residual network in Hardt and Ma [2016]), an even stronger conclusion is possible: all critical points of the $[\lambda = 0]$ problem are global minimum. In experiments, some of these properties are also empirically observed in deep nonlinear networks; cf., Choromanska et al. [2015b], Dauphin et al. [2014], Saxe et al. [2013].

In this paper, we consider the following regularized form of the optimization problem:

$$\begin{aligned} \underset{A}{\text{Minimize:}} \quad & \mathsf{J}[A] = \mathsf{E}[\, \lambda \int_0^T \frac{1}{2}\mathrm{tr}\,(A_t^\top \, A_t)\,\mathrm{d}t \; + \; \frac{1}{2}|X_T - Z|^2 \,] \\ \text{Subject to:} \quad & \frac{\mathrm{d}X_t}{\mathrm{d}t} = A_t X_t, \quad X_0 \sim p_0 \end{aligned} \tag{4}$$

where $\lambda \in \mathbb{R}^+ := \{x \in \mathbb{R} \; : \; x \geq 0\}$ is a regularization parameter. In literature, this form of regularization is referred to as weight decay [Goodfellow et al., 2016, Sec. 7.1.1]. Eq. (4) is an example of an optimal control problem and is referred to as such. The limit $\lambda \downarrow 0$ is referred to as $[\lambda = 0^+]$ problem. The symbol $\mathrm{tr}(\cdot)$ and superscript $^\top$ are used to denote matrix trace and matrix transpose, respectively.

The regularized problem is important because of the following reasons:

(i) The learning algorithms are believed to converge to the critical points of the regularized $[\lambda = 0^+]$ problem, a phenomenon known as implicit regularization Neyshabur et al. [2014], Zhang et al. [2016], Gunasekar et al. [2017].

(ii) It is shown in the paper that the stochastic gradient descent (for the functional J) yields the following learning algorithm for the weights $A_t$:

$$A_t^{(k+1)} = A_t^{(k)} + \eta_k(-\lambda A_t^{(k)} + \text{backprop update}) \tag{5}$$

for $k = 1, 2, \ldots$, where $\eta_k$ is the learning rate parameter. Thus, the parameter $\lambda$ models dissipation (or weight decay) in backprop. In an implementation of backprop, one would expect to obtain critical points of the $[\lambda = 0^+]$ problem.

The outline of the remainder of this paper is as follows: The Hamilton's formulation is introduced for the optimal control problem (4) in Sec. 2; cf., LeCun et al. [1988], Farotimi et al. [1991] for related constructions. The Hamilton's equations are used to obtain a formula for the gradient of J, and subsequently derive the stochastic gradient descent learning algorithm of the form (5). The equations for the critical points of J are obtained by applying the Maximum Principle of optimal control (Prop. 1). Remarkably, the Hamilton's equations for the critical points can be solved in closed-form to obtain a characterization of the critical points in terms of the solutions of a nonlinear matrix-valued equation, referred to as the characteristic equation (Prop. 2). For a certain special case, where the matrix $R$ is normal, analytical results are obtained based on the use of the implicit function theorem (Thm. 2). Numerical continuation is employed to compute the solutions for this and the more general non-normal cases (Examples 1 and 2).

## 2 Hamilton's formulation and the learning algorithm

**Definition 1.** *The control Hamiltonian is the function*

$$\mathsf{H}(x, y, B) = y^\top B x - \frac{\lambda}{2} tr(B^\top B) \tag{6}$$

*where $x \in \mathbb{R}^d$ is the state, $y \in \mathbb{R}^d$ is the co-state, and $B \in M_d(\mathbb{R})$ is the weight matrix. The partial derivatives are denoted as $\frac{\partial \mathsf{H}}{\partial x}(x, y, B) := B^\top y$, $\frac{\partial \mathsf{H}}{\partial y}(x, y, B) := Bx$, and $\frac{\partial \mathsf{H}}{\partial B}(x, y, B) := yx^\top - \lambda B$.*

Pontryagin's Maximum Principle (MP) is used to obtain the Hamilton's equations for the solution of the optimal control problem (4). The MP represents a necessary condition satisfied by any minimizer. Conversely, a solution of the Hamilton's equation is a critical point of the functional J. The proof of the following proposition appears in the supplementary material.

**Proposition 1.** *Consider the terminal cost optimal control problem (4) with $\lambda \geq 0$. Suppose $A_t$ is the minimizer and $X_t$ is the corresponding trajectory. Then there exists a random process $Y : [0, T] \to \mathbb{R}^d$ such that*

$$\frac{\mathrm{d}X_t}{\mathrm{d}t} = +\frac{\partial \mathsf{H}}{\partial y}(X_t, Y_t, A_t) = +A_t X_t, \quad X_0 \sim p_0 \tag{7}$$

$$\frac{\mathrm{d}Y_t}{\mathrm{d}t} = -\frac{\partial \mathsf{H}}{\partial x}(X_t, Y_t, A_t) = -A_t^\top Y_t, \quad Y_T = Z - X_T \tag{8}$$

*and $A_t$ maximizes the expected value of the Hamiltonian*

$$A_t = \underset{B \in M_d(\mathbb{R})}{\arg\max} \quad \mathsf{E}[\mathsf{H}(X_t, Y_t, B)] \overset{(\lambda > 0)}{=} \frac{1}{\lambda} \mathsf{E}[Y_t X_t^\top] \tag{9}$$

*Conversely, if there exists $A_t$ and the pair $(X_t, Y_t)$ such that equations (7)-(8)-(9) are satisfied, then $A_t$ is a critical point of the optimization problem (4).*

**Remark 1.** The Maximum Principle can also be used to derive analogous (difference) equations in discrete-time as well as nonlinear settings. It is equivalent to the method of Lagrange multipliers that is used to derive the backprop algorithm in MNN, e.g., LeCun et al. [1988]. The continuous-time limit is considered here because the computations are simpler and the results are more insightful. Similar considerations have also motivated the study of continuous-time limit of other types of optimization algorithms, e.g., Su et al. [2014], Wibisono et al. [2016].

The Hamiltonian is also used to express the first order variation in the functional J. For this purpose, define the Hilbert space of matrix-valued functions $L^2([0,T]; M_d(\mathbb{R})) := \{A : [0,T] \to M_d(\mathbb{R}) \mid \int_0^T \mathrm{tr}(A_t^\top A_t)\, \mathrm{d}t < \infty\}$ with the inner product $\langle A, V \rangle_{L^2} := \int_0^T \mathrm{tr}(A_t^\top V_t)\, \mathrm{d}t$. For any $A \in L^2$, the gradient of the functional J evaluated at $A$ is denoted as $\nabla \mathsf{J}[A] \in L^2$. It is defined using the directional derivative formula:

$$\langle \nabla \mathsf{J}[A], V \rangle_{L^2} := \lim_{\epsilon \to 0} \frac{\mathsf{J}(A + \epsilon V) - \mathsf{J}(A)}{\epsilon}$$

where $V \in L^2$ prescribes the direction (variation) along which the derivative is being computed. The explicit formula for $\nabla \mathsf{J}$ is given by

$$\nabla \mathsf{J}[A] := -\mathsf{E}\left[\frac{\partial \mathsf{H}}{\partial B}(X_t, Y_t, A_t)\right] = \lambda A_t - \mathsf{E}\left[Y_t X_t^\top\right] \tag{10}$$

where $X_t$ and $Y_t$ are the obtained by solving the Hamilton's equations (7)-(8) with the prescribed (not necessarily optimal) weight matrix $A \in L^2$. The significance of the formula is that the steepest descent in the objective function J is obtained by moving in the direction of the steepest (for each fixed $t \in [0, T]$) ascent in the Hamiltonian H. Consequently, a stochastic gradient descent algorithm to learn the weights is as follows:

$$A_t^{(k+1)} = A_t^{(k)} - \eta_k(\lambda A_t^{(k)} - Y_t^{(k)} X_t^{(k)^\top}), \tag{11}$$

where $\eta_k$ is the step-size at iteration $k$ and $X_t^{(k)}$ and $Y_t^{(k)}$ are obtained by solving the Hamilton's equations (7)-(8):

$$\text{(Forward propagation)} \quad \frac{\mathrm{d}}{\mathrm{d}t} X_t^{(k)} = +A_t^{(k)} X_t^{(k)}, \quad \text{with init. cond. } X_0^{(k)} \tag{12}$$

$$\text{(Backward propagation)} \quad \frac{\mathrm{d}}{\mathrm{d}t} Y_t^{(k)} = -A_t^{(k)\top} Y_t^{(k)}, \quad Y_T^{(k)} = \underbrace{Z^{(k)} - X_T^{(k)}}_{\text{error}} \tag{13}$$

based on the sample input-output $(X^{(k)}, Z^{(k)})$. Note the forward-backward structure of the algorithm: In the forward pass, the network output $X_T^{(k)}$ is obtained given the input $X_0^{(k)}$; In the backward pass, the error between the network output $X_T^{(k)}$ and true output $Z^{(k)}$ is computed and propagated backwards. The regularization parameter is also interpreted as the dissipation or the weight decay parameter. By setting $\lambda = 0$, the standard backprop algorithm is obtained. A convergence result for the learning algorithm for the $[\lambda = 0]$ case appears as part of the supplementary material.

In the remainder of this paper, the focus is on the analysis of the critical points.

## 3  Critical points

For continuous-time networks, the critical points of the $[\lambda = 0]$ problem are all global minimizers (An analogous result for residual MNN appears in [Hardt and Ma, 2016, Thm. 2.3]).

**Theorem 1.** *Consider the $[\lambda = 0]$ optimization problem (4) with non-singular $\Sigma_0$. For this problem (provided a minimizer exists) every critical point is a global minimizer. That is,*

$$\nabla \mathsf{J}[A] = 0 \quad \Longleftrightarrow \quad \mathsf{J}(A) = \mathsf{J}^* := \min_A \mathsf{J}[A]$$

*Moreover, for any given (not necessarily optimal) $A \in L^2$,*

$$\|\nabla \mathsf{J}[A]\|_{L^2}^2 \geq T \, e^{-2 \int_0^T \sqrt{tr(A_t^\top A_t)} \, \mathrm{d}t} \, \lambda_{min}(\Sigma_0)(\mathsf{J}(A) - \mathsf{J}^*) \tag{14}$$

*where $\lambda_{min}(\Sigma_0)$ is the smallest eigenvalue of $\Sigma_0$.*

*Proof.* (Sketch) For the linear system (2), the fundamental solution matrix is denoted as $\phi_{t;t_0}$. The solutions of the Hamilton's equations (7)-(8) are given by

$$X_t = \phi_{t;0} X_0, \quad Y_t = \phi_{T;t}^\top (Z - X_T)$$

Using the formula (10) upon taking an expectation

$$\nabla \mathsf{J}[A] = -\phi_{T;t}^\top (R - \phi_{T;0}) \Sigma_0 \phi_{t;0}^\top$$

which (because $\phi$ is invertible) proves that:

$$\nabla \mathsf{J}[A] = 0 \quad \Longleftrightarrow \quad \phi_{T;0} = R \quad \Longleftrightarrow \quad \mathsf{J}(A) = \mathsf{J}^* := \min_A \mathsf{J}[A]$$

The derivation of the bound (14) is equally straightforward and appears as part of the supplementary material. □

Although the result is attractive, the conclusion is somewhat misleading because (as we will demonstrate with examples) even a small amount of regularization can lead to local (but not global) minimum as well as saddle point solutions.

**Assumption:** The following assumption is made throughout the remainder of this paper:

(i) **Property P1:** The matrix $R$ has no eigenvalues on $\mathbb{R}^- := \{x \in \mathbb{R} \ : \ x \leq 0\}$. The matrix $R$ is non-derogatory. That is, no eigenvalue of $R$ appears in more than one Jordan block.

For the scalar ($d = 1$) case, this property means $R$ is strictly positive. For the scalar case, the fundamental solution is given by the closed form formula $\phi_{T,0} = e^{\int_0^T A_t \, \mathrm{d}t}$. Thus, the positivity of $R$ is seen to be necessary to obtain a meaningful solution.

For the vector case, this property represents a sufficient condition such that $\log(R)$ can be defined as a real-valued matrix. That is, under property (P1), there exists a (not necessarily unique[2]) matrix $\log(R) \in M_d(\mathbb{R})$ whose matrix exponential $e^{\log(R)} = R$; cf., Culver [1966], Higham [2014]. The logarithm is trivially a minimum for the $[\lambda = 0]$ problem. Indeed, $A_t \equiv \frac{1}{T} \log(R)$ gives $X_t = e^{\frac{\log(R)}{T} t} X_0$ and thus $X_T = e^{\log(R)} X_0 = R X_0$. This shows $A_t$ can be made arbitrarily small by choosing a large enough depth $T$ of the network. An analogous result for the linear residual MNN appears in [Hardt and Ma, 2016, Thm. 2.1]. The question then is whether the constant solution $A_t \equiv \frac{1}{T} \log(R)$ is also obtained as a critical point for the $[\lambda = 0^+]$ problem?

The following proposition provides a complete characterization of the critical points (for the general $\lambda \in \mathbb{R}^+$ problem) in terms of the solutions of a matrix-valued characteristic equation:

**Proposition 2.** *The general solution of the Hamilton's equations* (7)-(9) *is given by*

$$X_t = e^{2t\Omega} \, e^{t\mathsf{C}^\top} X_0 \tag{15}$$

$$Y_t = e^{2t\Omega} \, e^{(T-t)\mathsf{C}} \, e^{-2T\Omega} \, (Z - X_T) \tag{16}$$

$$A_t = e^{2t\Omega} \mathsf{C} e^{-2t\Omega} \tag{17}$$

*where* $C \in M_d(\mathbb{R})$ *is an arbitrary solution of the characteristic equation*

$$\lambda C = F^\top (R - F)\Sigma_0 \tag{18}$$

*where* $F := e^{2T\Omega} e^{TC^\top}$ *and the matrix* $\Omega := \frac{1}{2}(C - C^\top)$ *is the skew-symmetric component of* $C$. *The associated cost is given by*

$$J[A] = \frac{\lambda T}{2} tr\left(C^\top C\right) + \frac{1}{2} tr\left((F - R)^\top (F - R)\Sigma_0\right) + \frac{1}{2} E[|\xi|^2]$$

*And the following holds:*

$$A_t \equiv C \iff C \text{ is normal } \overset{(\Sigma_0 = I)}{\implies} R \text{ is normal}$$

*Proof.* (Sketch) Differentiating both sides of (9) with respect to $t$ and using the Hamilton's equations (7)-(8), one obtains

$$\frac{dA_t}{dt} = -A_t^\top A_t + A_t A_t^\top$$

whose general solution is given by (17). The remainder of the analysis is straightforward and appears as part of the supplementary material. □

**Remark 2.** Prop. 2 shows that the answer to the question posed above concerning the constant solution $A_t \equiv \frac{1}{T} \log(R)$ is false in general for the $[\lambda = 0^+]$ problem: For $\lambda > 0$ and $\Sigma_0 = I$, a constant solution is a critical point only if $R$ is a normal matrix. For the generic case of non-normal $R$, any critical point is necessarily non-constant for any positive choice of the parameter $\lambda$. Some of these non-constant critical points are described as part of the Example 2.

**Remark 3.** The linear structure of the input-output model (1) is not necessary to derive the results in Prop. 2. For correlated input-output random variables $(X, Z)$, the general form of the characteristic equation is as follows:

$$\lambda C = F^\top (E[ZX_0^\top] - F\Sigma_0)$$

where (as before) $\Sigma_0 = E[X_0 X_0^\top]$, and $F := e^{2T\Omega} e^{TC^\top}$ where $\Omega := \frac{1}{2}(C - C^\top)$.

Prop. 2 is useful because it helps reduce the infinite-dimensional problem to a finite-dimensional characteristic equation (18). The solutions $C$ of the characteristic equation fully parametrize the solutions of the Hamilton's equations (7)-(9) which in turn represent the critical points of the optimal control problem (4).

The matrix-valued nonlinear characteristic equation (18) is still formidable. To gain analytical and numerical insight into the matrix case, the following strategy is employed:

(i) A solution $C$ is obtained by setting $\lambda = 0$ in the characteristic equation. The corresponding equation is

$$e^{T(C-C^\top)} e^{TC^\top} = R$$

This solution is denoted as $C(0)$.

(ii) Implicit function theorem is used to establish (local) existence of a solution branch $C(\lambda)$ in a neighborhood of the $\lambda = 0$ solution.

(iii) Numerical continuation is used to compute the solution $C(\lambda)$ as a function of the parameter $\lambda$.

The following theorem provides a characterization of normal solutions $C$ for the case where $R$ is assumed to be a normal matrix and $\Sigma = I$. Its proof appears as part of the supplementary material.

**Theorem 2.** *Consider the characteristic equation* (18) *where $R$ is assumed to be a normal matrix that satisfies the Property (P1) and $\Sigma_0 = I$.*

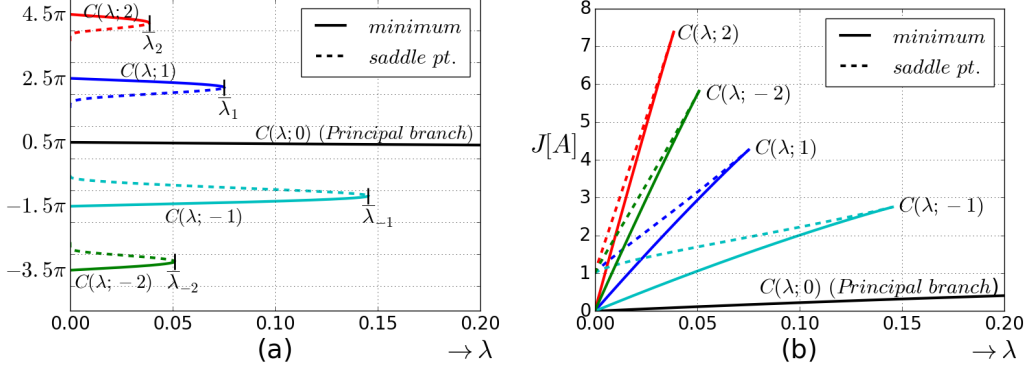

Figure 1: (a) Critical points in Example 1 (the $(2, 1)$ entry of the solution matrix $C(\lambda; n)$ is depicted for $n = 0, \pm 1, \pm 2$); (b) The cost $J[A]$ for these solutions.

(i) *For $\lambda = 0$ the normal solutions of (18) are given by $\frac{1}{T}\log(R)$.*

(ii) *For each such solution, there exists a neighborhood $\mathcal{N} \subset \mathbb{R}^+$ of $\lambda = 0$ such that the solution of the characteristic equation (18) is well-defined as a continuous map from $\lambda \in \mathcal{N} \to \mathsf{C}(\lambda) \in M_d(\mathbb{R})$ with $\mathsf{C}(0) = \frac{1}{T}\log(R)$. This solution is given by the asymptotic formula*

$$\mathsf{C}(\lambda) = \frac{1}{T}\log(R) - \frac{\lambda}{T^2}(RR^\top)^{-1}\log(R) + O(\lambda^2)$$

**Remark 4.** For the scalar case $\log(\cdot)$ is a single-valued function. Therefore, $A_t \equiv \mathsf{C} = \frac{1}{T}\log(R)$ is the unique critical point (minimizer) for the $[\lambda = 0^+]$ problem. While the $[\lambda = 0^+]$ problem admits a unique minimizer, the $[\lambda = 0]$ problem does not. In fact, any $A_t$ of the form $A_t = \frac{1}{T}\log(R) + \tilde{A}_t$ where $\int_0^T \tilde{A}_t \, dt = 0$ is also a minimizer of the $[\lambda = 0]$ problem. So, while there are infinitely many minimizers of the $[\lambda = 0]$ problem, only one of these survives with even a small amount of regularization. A global characterization of critical points as a function of parameters $(\lambda, R, \Sigma_0, T) \in \mathbb{R}^+ \times \mathbb{R}^+ \times \mathbb{R}^+ \times \mathbb{R}^+$ is possible and appears as part of the supplementary material.

**Example 1** (Normal matrix case)**.** Consider the characteristic equation (18) with $R = \begin{bmatrix} 0 & -1 \\ 1 & 0 \end{bmatrix}$ (rotation in the plane by $\pi/2$), $\Sigma_0 = I$ and $T = 1$. For $\lambda = 0$, the normal solutions of the characteristic equation are given by the multi-valued matrix logarithm function:

$$\log(R) = (\pi/2 + 2n\pi)\begin{bmatrix} 0 & -1 \\ 1 & 0 \end{bmatrix} =: \mathsf{C}(0; n), \quad n = 0, \pm 1, \pm 2, \dots$$

It is easy to verify that $e^{\mathsf{C}(0;n)} = R$. $\mathsf{C}(0; 0)$ is referred to as the principal logarithm.

The software package PyDSTool Clewley et al. [2007] is used to numerically continue the solution $C(\lambda; n)$ as a function of the parameter $\lambda$. Fig. 1(a) depicts the solutions branches in terms of the $(2, 1)$ entry of the matrix $C(\lambda; n)$ for $n = 0, \pm 1, \pm 2$. The following observations are made concerning these solutions:

(i) For each fixed $n \neq 0$, there exist a range $(0, \bar{\lambda}_n)$ for which there exist two solutions, a local minimum and a saddle point. At the limit (turning) point $\lambda = \bar{\lambda}_n$, there is a qualitative change in the solution from a minimum to a saddle point.

(ii) As a function of $n$, $\bar{\lambda}_n$ decreases monotonically as $|n|$ increases. For $\lambda > \bar{\lambda}_{-1}$, only a single solution, the principal branch $C(\lambda; 0)$ was found using numerical continuation.

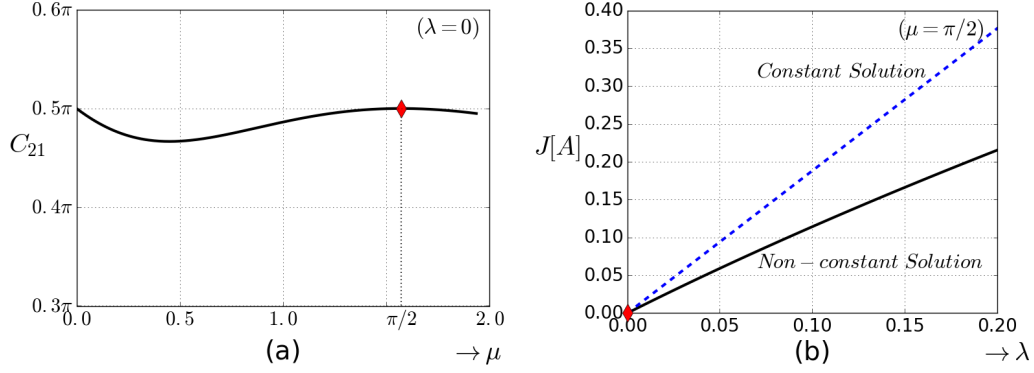

Figure 2: (a) Numerical continuation of the solution in Example 2; (b) The cost $J[A]$ for the critical point (minimum) and the constant $\frac{1}{T}\log(R)$ solution.

(iii) Along the branch with a fixed $n \neq 0$, as $\lambda \downarrow 0$, the saddle point solution escapes to infinity. That is as $\lambda \downarrow 0$, the saddle point solution $C(\lambda; n) \to (\pi/2 + (2n-1)\pi) \begin{bmatrix} -\infty & -1 \\ 1 & -\infty \end{bmatrix}$. The associated cost $J[A] \downarrow 1$ (The cost of global minimizer $J^* = 0$).

(iv) Among the numerically obtained solution branches, the principal branch $C(\lambda; 0)$ has the lowest cost. Fig. 1 (b) depicts the cost for the solutions depicted in Fig. 1 (a).

The numerical calculations indicate that while the $[\lambda = 0]$ problem has infinitely many critical points (all global minimizers), only a finitely many critical points persist for any finite positive value of $\lambda$. Moreover, there exists both local (but not global) minimum as well as saddle points for this case. Among the solutions computed, the principal branch (continued from the principal logarithm $C(0; 0)$) has the minimum cost.

**Example 2** (Non-normal matrix case). Numerical continuation is used to obtain solutions for non-normal $R = \begin{bmatrix} 0 & -1 \\ 1 & \mu \end{bmatrix}$, where $\mu$ is a continuation parameter and $T = 1$. Fig. 2(a) depicts a solution branch as a function of parameter $\mu$. The solution is initialized with the normal solution $\mathsf{C}(0; 0)$ described in Example 1. By varying $\mu$, the solution is continued to $\mu = \pi/2$ (indicated as $\diamond$ in part (a)). This way, the solution $\mathsf{C} = \begin{bmatrix} 0 & 0 \\ \frac{\pi}{2} & 0 \end{bmatrix}$ is found for $R = \begin{bmatrix} 0 & -1 \\ 1 & \frac{\pi}{2} \end{bmatrix}$. It is easy to verify that $\mathsf{C}$ is a solution of the characteristic equation (18) for $\lambda = 0$ and $T = 1$. For this solution, the critical point of the optimal control problem $A_t = \begin{bmatrix} -\pi \sin(\pi t) & \pi \cos(\pi t) - \pi \\ \pi \cos(\pi t) + \pi & \pi \sin(\pi t) \end{bmatrix}$ is non-constant. The principal logarithm $\log(R) = \begin{bmatrix} -\gamma \tan \gamma & -\gamma \sec \gamma \\ \gamma \sec \gamma & \gamma \tan \gamma \end{bmatrix}$, where $\gamma = \sin^{-1}\left(\frac{\pi}{4}\right)$. The regularization cost for the non-constant solution $A_t$ is strictly smaller than the constant $\frac{1}{T}\log(R)$ solution:

$$\int_0^1 \operatorname{tr}(A_t A_t^\top)\,dt = \int_0^1 \operatorname{tr}(\mathsf{C}\mathsf{C}^\top)\,dt = \frac{\pi^2}{4} < 3.76 = \int_0^1 \operatorname{tr}(\log(R)\log(R)^\top)\,dt$$

Next, the parameter $\mu = \frac{\pi}{2}$ is fixed, and the solution continued in the parameter $\lambda$. Fig. 2(b) depicts the cost $J[A]$ for the resulting solution branch of critical points (minimum). The cost with the constant $\frac{1}{T}\log(R)$ is also depicted. It is noted that the latter is not a critical point of the optimal control problem for any positive value of $\lambda$.

# 4 Conclusions and directions for future work

In this paper, we studied the optimization problem of learning the weights of a linear neural network with mean-squared loss function. In order to do so, we introduced a novel formulation:

 (i) The linear network is modeled as a continuous time (surrogate for layer) optimal control problem;

(ii) A weight decay type regularization is considered where the interest is in the limit as the regularization parameter $\lambda \downarrow 0$ (the limit is referred to as the $[\lambda = 0^+]$ problem).

The Maximum Principle of optimal control theory is used to derive the Hamilton's equations for the critical points. A remarkable result of our paper is that the critical point solutions of the infinite-dimensional problem are completely characterized via the solutions of a finite-dimensional characteristic equation (Eq. (18)). That such a reduction is possible is unexpected because the weight update equation is nonlinear (even in the settings of linear networks).

Based on the analysis of the characteristic equation, several conclusions are obtained[3]:

 (i) It has been noted in literature that, for linear networks, *all critical points are global minimum*. While this is also true here for the $[\lambda = 0]$ and the $[\lambda = 0^+]$ problems, even a small amount of regularization alters the picture, e.g., saddle points emerge (Example 1).

(ii) The critical points of the regularized $[\lambda = 0^+]$ problem is qualitatively very different compared to the non-regularized $[\lambda = 0]$ problem (Remark 4). Several quantitative results on the critical points of the regularized problem are described in Theorem 2 and Examples 1 and 2.

(iii) The study of the characteristic equation revealed an unexpected qualitative difference in the critical points between the two cases where $R := \mathsf{E}[ZX_0^\top]$ is a normal or non-normal matrix. In the latter (generic) case, the network weights are necessarily non-constant (Prop. 2).

We believe that the ideas and tools introduced in this paper will be useful for the researchers working on the analysis of deep learning. In particular, the paper is expected to highlight and spur work on implicit regularization. Some directions for future work are briefly noted next:

 (i) **Non-normal solutions of the characteristic equation**: Analysis of the non-normal solutions of the characteristic equation remains an open problem. The non-normal solutions are important because of the following empirical observation (summarized as part of the supplementary material): In numerical experiments with learning, the weights can get stuck at non-normal critical points before eventually converging to a "good" minimum.

(ii) **Generalization error**: With a finite number of samples $(X_0^i, Z^i)_{i=1}^N$, the characteristic equation
$$\lambda C = F^\top (R - F)\Sigma_0^{(N)} + F^\top Q^{(N)}$$
where $\Sigma_0^{(N)} := \frac{1}{N} \sum_{i=1}^N X_0^i X_0^{i\top}$ and $Q^{(N)} := \frac{1}{N} \sum_{i=1}^N X_0^i \xi^{i\top}$. Sensitivity analysis of the solution of the characteristic equation, with respect to variations in $\Sigma_0^{(N)}$ and $Q^{(N)}$, can shed light on the generalization error for different critical points.

(iii) **Second order analysis**: The paper does not contain second order analysis of the critical points – to determine whether they are local minimum or saddle points. Based on certain preliminary results for the scalar case, it is conjectured that the second order analysis is possible in terms of the first order variation for the characteristic equation.

## Footnotes

[2]Under Property (P1), $\log(R)$ is uniquely defined if and only if all the eigenvalues of $R$ are positive. When not unique there are countably many matrix logarithms, all denoted as $\log(R)$. The principal logarithm of $R$ is the unique such matrix whose eigenvalues lie in the strip $\{z \in \mathbb{C} \ : \ -\pi < \mathrm{Im}(z) < \pi\}$.

[3]Qualitative aspects of some of the conclusions may be *obvious* to experts in Deep Learning. The objective here is to obtain quantitative characterization in the (relatively tractable) setting of linear networks.

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
