[Supplementary Material]



## I. APPENDIX

**Notation:** For all $B \in M_d(\mathbb{R})$, the Frobenius norm is denoted as $\|B\|$ given by $\|B\| := \sqrt{\mathrm{tr}(BB^\top)}$.

### A. Scalar case

The scalar case is analyzed using elementary means and is useful to both introduce the characteristic equation as well as highlight the difference between the $[\lambda = 0]$ and the $[\lambda = 0^+]$ problems.

**Theorem 1.** *Consider the terminal cost optimal control problem* (4) *for the scalar* $(d = 1)$ *case with* $R > 0$ *and* $\Sigma_0 = \mathsf{E}[X_0^2] > 0$ *given. If* $A_t$ *is a minimizer then*

$$A_t \equiv \mathsf{C}, \quad X_t = e^{t\mathsf{C}}X_0 \tag{I.1}$$

*where the constant* $\mathsf{C}$ *is a solution of the characteristic equation*

$$\lambda \mathsf{C} = e^{T\mathsf{C}}(R - e^{T\mathsf{C}})\Sigma_0 \tag{I.2}$$

*Conversely a solution* $\mathsf{C}$ *of the characteristic equation* (I.2) *defines a critical point* (I.1) *of the optimal control problem* (4).

*The following is a complete characterization of the solutions* $\mathsf{C}$ *of the characteristic equation* (I.2) *as a function of parameters* $(\lambda, R, \Sigma_0, T) \in \mathbb{R}^+ \times \mathbb{R}^+ \times \mathbb{R}^+ \times \mathbb{R}^+$:

  (i) *For* $\lambda \in [0, 2e^3\Sigma_0 T]$ *there exists a unique solution. The associated solution obtained using* (I.1) *is a minimizer.*

  (ii) *In the asymptotic limit as* $\lambda \downarrow 0$, *the minimizer is given by an asymptotic expansion*

$$\mathsf{C} = \frac{1}{T}\log(R) - \lambda \frac{\log(R)}{T^2 R^2 \Sigma_0} + O(\lambda^2) \tag{I.3}$$

  *The unique solution for the* $\lambda = 0^+$ *problem, obtained by retaining the first order term, is given by* $\mathsf{C} = \frac{1}{T}\log(R)$.

  (iii) *For* $\lambda > 2e^3\Sigma_0 T$, *there exists an interval such that for* $R \in [R_1(\lambda), R_2(\lambda)]$ *there are exactly 3 solutions of the characteristic equation. For* $R > R_2(\lambda)$ *or* $R < R_1(\lambda)$ *there exists exactly one solution.*

*Proof.* In the scalar case, the state is given by the explicit formula $X_t = e^{\int_0^t A_s \, ds}X_0$. Therefore, the objective function

$$\mathsf{J}[A] = \lambda \int_0^T A_t^2 \, dt + (e^{\int_0^T A_t \, dt} - R)^2 \Sigma_0$$

Using the Jensen's inequality

$$\mathsf{J}[A] \geq \frac{\lambda}{T}(\int_0^T A_t \, dt)^2 + (e^{\int_0^T A_t \, dt} - R)^2 \Sigma$$

with an equality iff $A_t \equiv C$, a constant. Therefore

$$\min_{A \in L^2} J(A) = \min_{C \in \mathbb{R}} \lambda T C^2 + (e^{TC} - R)^2 \Sigma$$

The characteristic equation is the first order optimality condition of the right hand side.

(i) Denote $\tilde{\lambda} = \frac{\lambda}{T\Sigma}$ and $\tilde{C} = TC$ to write the characteristic equation as

$$f(\tilde{C}, \tilde{\lambda}) := \tilde{\lambda} \tilde{C} e^{-\tilde{C}} + e^{\tilde{C}} = R \tag{I.4}$$

For $\tilde{\lambda} = 0$, the solution $\tilde{C} = \log(R)$. For $\lambda > 0$, $f$ is onto (since $f$ is continuous and $\lim_{\tilde{C} \to \pm\infty} f(\tilde{C}) = \pm\infty$). Therefore, there exists at least one solution for each given $R$. Since $f'(\tilde{C}) = \lambda e^{-\tilde{C}}(1 - \tilde{C}) + e^{\tilde{C}} > 0$ for $\tilde{C} \le 1$, $f$ is monotone on $(-\infty, 1]$. Also $f'(\tilde{C}) = 0 \Leftrightarrow \lambda = \frac{e^{2\tilde{C}}}{\tilde{C}-1}$ and $\frac{e^{2\tilde{C}}}{\tilde{C}-1}$ is a unimodal convex function for $\tilde{C} > 1$ with minimum $2e^3$ at $\tilde{C} = 3/2$. Therefore for $\tilde{\lambda} \le 2e^3$, $f$ is monotone over entire $\mathbb{R}$. This implies that the solution to $f(\tilde{C}) = R$ is unique for $\tilde{\lambda} \le 2e^3$.

(ii) At $\tilde{\lambda} = 0$, $\tilde{C} = \log(R)$. Also $f'(\log(R), 0) = R \ne 0$. So by the implicit function theorem there exists a unique solution $\tilde{\lambda} \to \tilde{C}(\tilde{\lambda})$ in a neighborhood of 0. The asymptotic formula (I.3) for the solution is obtained by substituting regular perturbation expansion $\tilde{C} = \tilde{C}_0 + \lambda \tilde{C}_1 + O(\lambda^2)$ into (I.4).

$$f(\tilde{C}) = \lambda \tilde{C}_0 e^{-\tilde{C}_0} + e^{\tilde{C}_0}(1 + \lambda \tilde{C}_1) + O(\lambda^2) = R$$

Collecting the zeroth and the first order terms, one obtains $\tilde{C}_0 = \log(R)$ and $\tilde{C}_1 = -\frac{\log(R)}{R^2}$.

(iii) If $\tilde{\lambda} > 2e^3$, $f'(\tilde{C}) = 0$ has two solutions, $\tilde{C}_1 \in (3/2, \infty)$ and $\tilde{C}_2 \in (1, 3/2)$. Therefore for $R \in [f(\tilde{C}_1), f(\tilde{C}_2)]$, $f(\tilde{C}) = R$ has three solutions.

$\square$

## B. Proof of the Proposition 1 (Hamiltonian formulation)

Let $A_t$ be the minimizer of (4). Define $X_t$ and $Y_t$ as the solutions of the Hamilton's equations (7)-(8). We show $A_t$ satisfies (9) as follows: For $s \in [0, T]$ and $B \in M_d(\mathbb{R})$ consider a (needle) variation of the form:

$$A_t^{(\varepsilon)} = \begin{cases} B & t \in [s - \varepsilon, s] \\ A_t & t \notin [s - \varepsilon, s] \end{cases}$$

Let $X_t^{(\varepsilon)}$ denote the solution to the Hamitonian equation-(7) with $A_t^{(\varepsilon)}$. It is given by:

$$X_t^{(\varepsilon)} = X_t + \varepsilon \eta_t + O(\varepsilon^2)$$

where for $t < s$, $\eta_t = 0$ and for $t > s$, $\eta_t$ is the solution of

$$\frac{d\eta_t}{dt} = A_t \eta_t, \quad \text{with i.c} \quad \eta_s = (B - A_s)X_s$$

The perturbed cost is

$$\mathsf{J}[A^{(\varepsilon)}] = \mathsf{J}[A] + \varepsilon\lambda\left(\mathrm{Tr}(B^\top B) - \mathrm{Tr}(A_s^\top A_s)\right) + 2\varepsilon\mathsf{E}[(X_T - Z)^\top \eta_T] + O(\varepsilon^2)$$

Since $A_t$ is a minimizer

$$\frac{\lambda}{2}\left(\mathrm{Tr}(B^\top B) - \mathrm{Tr}(A_s^\top A_s)\right) + \mathsf{E}[(X_T - Z)^\top \eta_T] \geq 0$$

The next step is to obtain $(X_T - Z)^\top \eta_T$ in terms of $Y_t$. By construction $\frac{\mathrm{d}}{\mathrm{d}t}Y_t^\top \eta_t = 0$. Therefore,

$$(Z - X_T)^\top \eta_T = Y_T^\top \eta_T = Y_s^\top \eta_s = Y_s^\top (B - A_s)X_s$$

and hence

$$\frac{\lambda}{2}\left(\mathrm{Tr}(B^\top B) - \mathrm{Tr}(A_s^\top A_s)\right) - \mathsf{E}[Y_s^\top (B - A_s)X_s] \geq 0,$$

On collecting the terms, one obtains

$$\mathsf{E}[H(X_s, Y_s, A_s)] \geq \mathsf{E}[H(X_s, Y_s, B)] \quad \forall B \in M_d(\mathbb{R})$$

Since $s \in [0, T]$ is arbitrary, the result follows. Conversely by Sec. I-C below, any weight matrix $A$ that satisfies $\nabla\mathsf{J}[A] = -\mathsf{E}[\frac{\partial H}{\partial B}(X_t, Y_t, A_t)] = 0$ is a critical point. The condition $\mathsf{E}[\frac{\partial H}{\partial B}(X_t, Y_t, A_t)] = 0$ is equivalent to equation (9) since $H$ is a concave function of $A$.

## C. First order variation of $\mathsf{J}$

Let $X_t$ and $Y_t$ be the solutions to the Hamilton's equations-(7)-(8) with weight matrix $A_t$. Define $A_t^{(\varepsilon)} = A_t + \varepsilon V_t$. Let $X_t^{(\varepsilon)}$ and $Y_t^{(\varepsilon)}$ be the solutions to the Hamilton's equations with weight matrix $A_t^{(\varepsilon)}$. In the limit as $\varepsilon \to 0$, $X_t^{(\varepsilon)}$ is given by the asymptotic formula $X_t^{(\varepsilon)} = X_t + \varepsilon\eta_t + O(\varepsilon^2)$ where

$$\frac{\mathrm{d}\eta_t}{\mathrm{d}t} = A_t\eta_t + V_t X_t, \quad \eta_0 = 0$$

In terms of $\eta_t$, the objective function

$$\mathsf{J}[A^{(\varepsilon)}] = \mathsf{J}[A] + \varepsilon\left(\lambda\int_0^T \mathrm{tr}(A_t^\top V_t) + \mathsf{E}[(X_T - Z)^\top \eta_T]\right) + O(\varepsilon^2)$$

Use the definition of $Y_t$ to express $(X_T - Z)^\top \eta_T$ as

$$(Z - X_T)^\top \eta_T = Y_T^\top \eta_T = \int_0^T \frac{\mathrm{d}}{\mathrm{d}t}(Y_t^\top \eta_t)\,\mathrm{d}t = \int_0^T \left(-Y_t^\top A_t\eta_t + Y_t^\top A_t\eta_t + Y_t^\top V_t X_t\right)\mathrm{d}t = \int_0^T Y_t^\top V_t X_t\,\mathrm{d}t$$

Therefore,

$$\mathsf{J}[A^{(\varepsilon)}] = \mathsf{J}[A] + \varepsilon\int_0^T \mathsf{E}\left[\mathrm{tr}\left((\lambda A_t^\top - X_t Y_t^\top)V_t\right)\right]\mathrm{d}t + O(\varepsilon^2)$$

On the other hand $\frac{\partial H}{\partial B}(x, y, B) = \lambda B + yx^\top$. Therefore

$$\mathsf{J}[A^{(\varepsilon)}] - \mathsf{J}[A] = \varepsilon\int_0^T \mathrm{tr}\left(\mathsf{E}\left[\frac{\partial H}{\partial B}(X_t, Y_t, A_t)\right]^\top V_t\right)\mathrm{d}t + O(\varepsilon^2)$$

which gives the result $\nabla\mathsf{J}[A] = -\mathsf{E}[\frac{\partial H}{\partial B}(X_t, Y_t, A_t)]$.

*D. Proof of the Theorem 1*

For the $[\lambda = 0]$ problem, the gradient $\nabla\mathsf{J}[A]$ is (by (10))

$$\nabla\mathsf{J}[A] = -\mathsf{E}[Y_t X_t^\top]$$

where $X_t$ and $Y_t$ solve the Hamilton's equations-(7)-(8). Define the state transition matrix $\phi(t,t_0)$ of the differential equation $\frac{\mathrm{d}X_t}{\mathrm{d}t} = A_t X_t$ according to:

$$\frac{\mathrm{d}\phi(t,t_0)}{\mathrm{d}t} = A_t \phi(t,t_0), \quad \phi(t_0,t_0) = I$$

In terms of the transition matrix,

$$X_t = \phi(t,0)X_0, \quad Y_t = \phi(T,t)^\top(Z - X_T)$$

Therefore

$$\nabla\mathsf{J}[A] = -\phi(T,t)^\top(R - \phi(T,0))\Sigma\phi(t,0)^\top =: \psi_t$$

Since $\phi(t,t_0)$ is invertible

$$\nabla\mathsf{J}[A] = 0 \quad \Leftrightarrow \quad R = \phi(T,0) \quad \Leftrightarrow \quad \mathsf{J}[A] = J^* = \mathsf{E}[|W|^2]$$

For each fixed $t \in [0,T]$

$$\begin{aligned}
\|\psi_t\|^2 &= \|\phi(T,t)^\top(R - \phi(T,0))\Sigma\phi(t,0)^\top\|^2 \\
&\geq \lambda_{\min}(\phi(T,t)^\top\phi(T,t))\lambda_{\min}(\phi(t,0)^\top\phi(t,0))\|(R - \phi(T,0))\Sigma\|^2 \\
&\geq \lambda_{\min}(\phi(T,t)^\top\phi(T,t))\lambda_{\min}(\phi(t,0)^\top\phi(t,0))\lambda_{\min}(\Sigma)\mathrm{tr}((R - \phi(T,0))^\top(R - \phi(T,0))\Sigma) \\
&\geq e^{-2\int_0^T \||A_t\|\,\mathrm{d}t}\lambda_{\min}(\Sigma)(\mathsf{J}[A] - \mathsf{J}^*)
\end{aligned}$$

where we used Lemma I.1 (see below) in the last step. Integrating the inequality on $[0,T]$ yields the result.

**Lemma I.1.** *Let $\phi(t,t_0)$ be the state transition matrix defined according to $\frac{\mathrm{d}}{\mathrm{d}t}\phi(t,t_0) = A_t\phi(t,t_0)$ with $\phi(t_0,t_0) = I$. Then,*

$$e^{-2\int_0^t \|A_t\|\,\mathrm{d}t} \leq \lambda_{min}(\phi(t,0)^\top\phi(t,0)) \leq \lambda_{max}(\phi(t,0)^\top\phi(t,0)) \leq e^{2\int_0^t \|A_t\|\,\mathrm{d}t}$$

*Proof.* Observe that

$$\lambda_{\max}(\phi(t,0)^\top\phi(t,0)) = \max_{x\neq 0}\frac{x^\top\phi_{t,0}^\top\phi_{t,0}x}{x^\top x} = \max_{x_0\neq 0}\frac{x_t^\top x_t}{x_0^\top x_0}$$

Now,

$$\frac{\mathrm{d}}{\mathrm{d}t}|x_t|^2 = x_t^\top(A_t + A_t^\top)x_t \leq \lambda_{\max}(A_t + A_t^\top)|x_t|^2 \leq 2\|A_t\||x_t|^2$$

where the last inequality follows because $2\|A_t\| = \|A_t + A_t^\top\| \geq \lambda_{\max}(A_t + A_t^\top)$. Therefore, $|x_t|^2 \leq e^{2\int_0^t \|A_t\|\,\mathrm{d}t}|x_0|^2$ which gives the upper bound. The calculation for the lower bound is similar. $\square$

## E. Convergence of the learning algorithm

**Proposition 1.** *Consider the stochastic gradient descent learning algorithm* (11) *with* $\lambda = 0$. *Suppose* $\exists \alpha > 0$ *such that* $\mathsf{E}[X_0 X_0^T Q X_0 X_0^T] \le \alpha \Sigma_0 Q \Sigma_0$ *for all symmetric matrices $Q$, and* $\|A^{(k)}\|_{L^2} \le M$ *for all* $k \in \mathbb{N}$. *Then there exists a positive constant* $\beta > 0$ *such that* $\mathsf{J}$ *is a $\beta$-smooth function. And for sufficiently small constant stepsize* $\eta_k = \eta \le \frac{1}{\alpha\beta}$,

$$\mathsf{J}[A^{(k)}] - \mathsf{J}^* \le (1 - \frac{\eta}{2})^k (\mathsf{J}[A^{(0)}] - \mathsf{J}^*) + \eta\beta e^{2M\sqrt{T}} \mathsf{E}[|X_0|^2 |\xi|^2],$$

*for all* $k \in \mathbb{N}$ *where* $\mathsf{J}^* := \min_A \mathsf{J}[A] = E[|\xi|^2]$.

*Proof.* The proof is based on Theorem 4.8 in [1] where it is shown that SGD converges to a local minimum. To apply the theorem we show

$$\mathsf{E}\left[ Y_t^{(k)} X_t^{(k)\top} \right] = \nabla\mathsf{J}[A^{(k)}]$$

because $X_0^{(k)}$ is a random sample of $X_0$ and $\nabla\mathsf{J}[A^{(K)}]$ is given by the formula (10) for $\lambda = 0$. Next

$$\mathsf{E}\left[ \left\| Y_t^{(k)} X_t^{(k)\top} \right\|^2 \right] = \mathsf{E}\left[ \|\phi(T,t)^\top (Z - X_T^i) X_0^{i\top} \phi(t,0)^\top\|^2 \right]$$

$$= \mathsf{E}\left[ \|\phi(T,t)^\top (R - \phi(T,0)) X_0 X_0^\top \phi(t,0)^\top\|^2 \right] + \mathsf{E}\left[ \|\phi(T,t)^\top W X_0^\top \phi(t,0)^\top\|^2 \right]$$

$$\le \alpha \|\phi(T,t)^\top (R - \phi(T,0)) \Sigma \phi(t,0)^\top\|^2 + e^{2\int_0^T \|A_t\| \mathrm{d}t} \mathsf{E}[|W|^2] \mathsf{E}[|X_0|^2]$$

where the assumption $\mathsf{E}[X_0 X_0^\top \phi(t,0)^\top \phi(t,0) X_0 X_0^\top] \le \alpha \Sigma \phi(t,0)^\top \phi(t,0) \Sigma$ and Lemma I.1 is used.

The fact that $\mathsf{J}$ is $\beta$-smooth is true since all the functions involved are smooth and it is assumed $A$ is bounded. Applying Theorem 4.8 in [1], SGD algorithm converges to a local minimum. The geometric convergence to the global minimum follows from Theorem 1 where it is shown that local minimum are global minimum for $\lambda = 0$ and using the inequality (14). $\square$

## F. Proof of Proposition 2

Suppose $(X_t, Y_t, A_t)$ is a solution of the Hamilton's equations(7)-(9). Then by differentiating $A_t$ with respect to $t$, one obtains

$$\frac{\mathrm{d}A_t}{\mathrm{d}t} = -A_t^\top A_t + A_t A_t^\top$$

On expressing $A_t = S_t + \Omega_t$ as the sum of its symmetric component $S_t = \frac{1}{2}(A_t + A_t^\top)$ and the skew-symmetric component $\Omega_t = \frac{1}{2}(A_t - A_t^\top)$, one obtains

$$\frac{\mathrm{d}S_t}{\mathrm{d}t} = 2\Omega_t S_t - 2S_t \Omega_t, \quad \frac{\mathrm{d}\Omega_t}{\mathrm{d}t} = 0$$

whose solution is given by

$$S_t = e^{2t\Omega}S_0 e^{-2t\Omega}, \quad \Omega_t = \Omega_0$$

This gives (17).

Using the formula (17) for $A_t$, the Hamilton's equation for $X_t$ is

$$\frac{\mathrm{d}X_t}{\mathrm{d}t} = e^{2t\Omega}Se^{-2t\Omega}X_t + \Omega X_t$$

whose solution is given by (15).

The optimal costate trajectory is obtained similarly. The Hamilton's equation for the costate is:

$$\frac{\mathrm{d}Y_t}{\mathrm{d}t} = -e^{2t\Omega}Se^{-2t\Omega}Y_t + \Omega Y_t, \quad Y_T = Z - X_T$$

whose solution is given by (16).

The characteristic equation (18) is obtained by using the formula $A_t = \frac{1}{\lambda}\mathsf{E}[Y_t X_t']$:

$$\lambda e^{2t\Omega}\mathsf{C}e^{-2t\Omega} = e^{2t\Omega}e^{-t\mathsf{C}}e^{T\mathsf{C}}e^{-2T\Omega}\mathsf{E}[(Z-X_T)X_0^\top]e^{t\mathsf{C}}e^{-2t\Omega}$$

upon multiplying both sides from left by $e^{t\mathsf{C}}e^{-2t\Omega}$ and from right by $e^{2t\Omega}e^{-t\mathsf{C}}$.

**Optimal cost:** Optimal cost is obtained by inserting $A_t = e^{2t\Omega}\mathsf{C}e^{-2t\Omega}$ into the cost function where the following identities are used:

$$\mathrm{tr}(A_t A_t^\top) = \mathrm{tr}(\mathsf{C}\mathsf{C}^\top)$$
$$\mathsf{E}[|X_T - Z|^2] = \mathsf{E}[|W|^2] + \mathsf{E}[|FX_0 - RX_0|^2]$$

**Constant $\Leftrightarrow$ normal:** Suppose $A_t = C$ a constant. Then $\frac{\mathrm{d}A_t}{\mathrm{d}t} = -A_t^\top A_t + A_t A_t^\top = 0$, and hence $A_t = C$ is a normal matrix. Conversely, assuming $A_t$ is a normal matrix implies $\frac{\mathrm{d}A_t}{\mathrm{d}t} = 0$ and hence $A_t = C$ a constant.

**Normal solution:** If $\mathsf{C}$ is normal, then $\mathsf{C}$ and $\Omega$ commute, therefore $F = e^{T\mathsf{C}}$. Hence the characteristic simplifies to

$$\lambda\mathsf{C} = e^{\mathsf{C}^\top}(R - e^{\mathsf{C}})\Sigma$$

and equivalently

$$\lambda\mathsf{C}e^{-\mathsf{C}^\top}\Sigma^{-1} + e^{\mathsf{C}} = R$$

Therefore, if $\mathsf{C}$ and $\Sigma$ commute (always true when $\Sigma = I$), $R$ is a normal matrix. We have proved

$$A_t \equiv \mathsf{C} \iff \mathsf{C} \text{ is normal} \overset{(\Sigma=I)}{\implies} R \text{ is normal}$$

Therefore a non-normal $R$ implies the minimizer $A_t$ is not constant for $\Sigma = I$.

*G. Proof of Theorem 2*

1) If $\mathsf{C}$ is normal, then $e^{T(\mathsf{C}-\mathsf{C}^\top)}e^{T\mathsf{C}} = e^{T\mathsf{C}}$. Hence for $\lambda = 0$ problem the characteristic equation becomes $e^{T\mathsf{C}} = R$ whose solution is $\mathsf{C} = \frac{1}{T}\log(R)$, interpreted as multi-valued matrix logarithm function (see [2]).

2) For $\lambda > 0$ and $\Sigma = I$ the characteristic equation is:

$$\lambda \mathsf{C} e^{-\mathsf{C}^\top} + e^{\mathsf{C}} = R$$

Since $\mathsf{C}$ is normal, $R$ must be normal and moreover there exists a unitary (complex) matrix $U$ such that $U^* R U = D$ where $D = \mathrm{diag}(r_1,\ldots,r_d)$ with $r_n \in \mathbb{C}$. Let $\mu_n \in \mathbb{C}$ be solution to the equation

$$\lambda \mu_n e^{-\mu_n^*} + e^{\mu_n} = r_n \tag{I.5}$$

for $n = 1,\ldots,d$. Then $\mathsf{C} = UGU^*$ where $G = \mathrm{diag}(\mu_1,\ldots,\mu_d)$ is the normal solution to the characteristic equation since

$$\lambda G e^{-G^*} + e^G = D \quad \Rightarrow \quad \lambda UG e^{-G^*} U^* + U e^G U^* = UDU^* \quad \Rightarrow \quad \lambda \mathsf{C} e^{\mathsf{C}^\top} + e^{\mathsf{C}} = R$$

It thus suffices to analyze solutions to the complex equation (I.5). Denoting $\mu_n = x + iy$ and $r_n = e^{a+i\theta}$ the complex equation (I.5) is written as two real equations:

$$f_1(x,y;\lambda) := \lambda x e^{-x}\cos(y) - \lambda y e^{-x}\sin(y) + e^x\cos(y) = e^a\cos(\theta)$$

$$f_2(x,y;\lambda) := \lambda x e^{-x}\sin(y) + \lambda y e^{-x}\cos(y) + e^x\sin(y) = e^a\sin(\theta)$$

At $\lambda = 0$, there are countability many solutions given by $x_0 = a$ and $y_0 = \theta + m2\pi$ for $m \in \mathbb{Z}$. The Jacobian

$$\mathsf{D}f(x_0,y_0;0) = \begin{bmatrix} \frac{\partial f_1}{\partial x}(x_0,y_0,0) & \frac{\partial f_1}{\partial y}(x_0,y_0,0) \\ \frac{\partial f_2}{\partial x}(x_0,y_0,0) & \frac{\partial f_2}{\partial y}(x_0,y_0,0) \end{bmatrix} = \begin{bmatrix} e^{x_0}\cos(y_0) & -e^{x_0}\sin(y_0) \\ e^{x_0}\sin(y_0) & e^{x_0}\cos(y_0) \end{bmatrix}$$

is nonsingular since $\det(\mathsf{D}f) = e^{2x_0} = e^{2a} > 0$. Therefore, using the implicit function theorem, there exists a neighborhood $\mathcal{N}$ of $\lambda = 0$ and a function $\lambda \in \mathcal{N} \to (x(\lambda),y(\lambda)) \in \mathbb{R}^2$ such that $f(x(\lambda),y(\lambda);\lambda) = 0$. The asymptotic formula for $x(\lambda)$ and $y(\lambda)$ are obtained upon using a regular perturbation expansion $x = x_0 + \lambda x_1 + O(\lambda^2)$ and $y = y_0 + \lambda y_1 + O(\lambda^2)$. Then

$$\begin{bmatrix} x_1 \\ y_1 \end{bmatrix} = -[\mathsf{D}f(x_0,y_0;0)]^{-1}\frac{\partial f}{\partial \lambda}(x_0,y_0;0)$$

$$= -e^{-x_0}\begin{bmatrix} \cos(y) & \sin(y_0) \\ -\sin(y) & \cos(y_0) \end{bmatrix}\begin{bmatrix} x_0 e^{-x_0}\cos(y_0) - y_0 e^{-x_0}\sin(y_0) \\ x_0 e^{-x_0}\sin(y_0) + y_0 e^{-x_0}\cos(y_0) \end{bmatrix}$$

$$= -e^{-2x_0}\begin{bmatrix} x_0 \\ y_0 \end{bmatrix}$$

Therefore

$$\mu = \log(r) - \lambda \frac{\log(r)}{|r|^2} + O(\lambda^2)$$

which gives the asymptotic formula

$$\mathsf{C} = \log(R) - \lambda (RR^*)^{-1} \log(R) + O(\lambda^2)$$

*H. Learning example*

The objective of this section is to present a numerical illustration of the learning algorithm (11). In particular we are interested in the behavior of the algorithm near critical points.

Let $R = \begin{bmatrix} 0 & -1 \\ 1 & 0 \end{bmatrix}$, $\Sigma_0 = I$, and $T = 1$ similar to the setting of Example 1. The regularization parameter $\lambda$ is set to 0.03. Weight matrix is initialized with the critical point corresponding to $\mathsf{C}(0.03, 2)$ perturbed with small Gaussian noise. 500 training samples are drawn from model 1, where $p_0$ is Gaussian $N(0, I_{2\times 2})$. The learning rate is $\eta = 0.05$ and at each iteration, one sample is selected randomly for learning. The test error is evaluated using 500 independent samples.

The test error versus number of iterations is depicted in Fig. 1 (a). It is observed that the test error remains approximately unchanged during a short interval at the beginning, and a longer interval afterwards. In these intervals, the weight matrix is close to a critical point corresponding to a non-normal solution of the characteristic equation. For example during the second interval, $A_t$ corresponds to the non-normal solution of the characteristic equation given by $\mathsf{C} = \begin{bmatrix} 0 & 0.5\pi \\ -2\pi & 0 \end{bmatrix}$. Eventually the algorithm converges to the global minimum which corresponds to the principal branch solution of the characteristic equation. The weight matrix at several instances of the algorithm is depicted in Fig. 1. Each curve is the $(2,1)$ entry of the weight matrix $A_t$ as a function of time.

Fig. 1: (a) The cost $J[A]$ (test error) versus iterations. (b) (2,1) entry of $A_t$ at iteration $k = 0, 1500, 15000, 25000$. The wight matrix $A_t$ at iteration $k = 1500, 15000$ correspond to non-normal solutions of the characteristic equation.