[Reviews · NeurIPS 2017]

Reviewer 1



The paper analyzes a particular setup of deep linear networks. Specifically, the depth is treated as a continuous variable, and the loss function is the mean squared error. The goal of the paper is to study the behavior of the critical points (number and type) as the regularization parameter changes. Due to the continuous depth setup, the authors can leverage tools from optimal control to characterize the solution of the problem. While the technical aspects of the paper are interesting, the conclusion of the paper seems to be less surprising (stated in lines 222-225), e.g. that the type of the critical points change as a function of lambda. This is expected without relying on such sophisticated analysis: at big lambda we get a convex loss, and thus the index of the critical point of the nonregularized function must eventually turn to zero (when the convex term dominates) from whatever value it had in the nonregularized function. Therefore, it is apparent that the changing lambda is able to affect the type of critical points. Please correct me if I am missing anything here. Despite my concern about the conclusion, I think the techniques used in the paper are interesting and could be useful for researchers who work on analysis of deep networks. In particular, the connection to optimal control via continuous depth idea is fascinating. A minor questions: Why Eq 11 and subsequently Eq 12 and Eq 13 do not depend on lambda? Lambda is still present in Eq 10.

Reviewer 2



The authors of this article consider an infinite-dimensional version of a deep linear network. They propose to learn the weights using an objective function that is the sum of a square loss and a Tikhonov regularizer, controlled by a regularization parameter lambda. The authors derive analytic characterizations for the critical points of the objective, and use them to numerically show that, even for arbitrarily small lambda, local minima and saddle points can appear, that do not exist when there is no regularization. I am not an expert of this subject, but it seems to me that looking at deep linear networks in an infinite-dimensional setting, instead of the usual finite-dimensional one, is a good idea. It seems to make computations simpler or at least more elegant, and could then allow for more in-depth analysis. Using methods from optimal control to study this problem is original, and apparently efficient. I also found the article well-written: it was pleasant to read, despite the fact that I am very unfamiliar with optimal control. However, I am not sure that the picture that the authors give of the critical points in the limit where lambda goes to zero is clear enough so that I understand what is surprising in it, and what algorithmic implications it can have. Indeed, when lambda is exactly zero, there are a lot of global minima. If these global minima can be grouped into distinct connected components, then, when lambda becomes positive, it is "normal" that local minima appear (intuitively, each connected component will typically give rise to at least one local minimum, but only one connected component - the one containing the global solution that minimizes the regularizer - will give rise to a local minimum that is also global). It seems to me that this is what the authors observe, in terms of local minima. The appearing of saddle points is maybe more surprising, but since they seem to come in "from infinity", they might not be an issue for optimization, at least for small values of lambda. Also, the authors focus on normal solutions. They are easier to compute than non-normal ones, but they are a priori not the only critical points (at least not when lambda=0). So maybe looking only at the normal critical points yields a very different "critical point diagram" than looking at all the critical points. If not, can the authors explain why? Finally, this is less important but, after all the formal computations (up to Theorem 2), I was slightly disappointed that the appearing of local minima and saddle points was only established for precise examples, and through numerical computations, and not via more rigorous arguments. Minor remarks: - It seems to me that the regularization proposed by the authors is essentially "weight decay", that is well-known among the people that train deep networks; it should be said. - Proposition 1: it is to me that it is not explicitly said that it only holds for positive values of lambda. - The covariance is defined as \Sigma_0 in the introduction but subsequently denoted by \Sigma. - l.129: it is said that the "critical points" will be characterized, but the proof uses Proposition 1, that deals with "minimizers", and not "critical points". - l. 141: "T" is missing in the characteristic equation. - Theorem 2 - (i): plural should probably be used, since there are several solutions. - Proof of theorem 2: the proof sketch seems to suggest that the authors assume with no proof that there always exists a normal solution when lambda > 0. The appendix actually proves the existence, but maybe the sketch could be slightly reformulated, so as not to give a wrong impression? - It seems to me that the authors never show that values of C that satisfy Equation (18) are critical points of (4) (the proposition says the opposite: critical points of (4) satisfy (18)). So how can they be sure that the solutions they compute by analytic continuation are indeed critical points of (4)? - I am not sure I understand the meaning of "robust" in the conclusion. Added after author feedback: thank you for the additional explanations. I am however slightly disappointed with your response to my main two comments. I agree that the emergence of saddle points, as well as the normal / non-normal difference are somewhat surprising. However, it seems to me that it is not really what the article presents as a surprise: if you think that these two points are the most surprising contribution of the article, it should probably be better highlighted and discussed. For the normal vs non-normal solutions, I do understand that studying non-normal critical points looks very difficult, and I am fine with it. However, I think that the abstract and introduction should then be clear about it, and not talk about general critical points, while the article only describes normal ones.

Reviewer 3



I have had some trouble with this paper as to be honest I am not familiar with this particular strand of the literature. i found the idea interesting though: Ie in effect, they study the properties of the learning dynamics by restating it as an optimal control problem, and then effectively use pertaining ideas (Pontryagin's maximum principle) to study equilibria (which exist due to the linearity of the problem). Some factors struck me as very peculiar though: - Why is it interesting to study linear NNs ? We already know that they have a unique optimal solution (as the authors seem to reconfirm ?) which moreover, we can find in closed-form. Furthermore, nonlinear NN might behave very differently, so studying the linear case might not tell us much about the interesting cases? The authors did not do a good job of explaining the motivation and should consider elaborating on these points. - I did not get the relevance of studying continuous networks (which do not exist). Could the authors instead appeal to discrete optimal control methods (why use 'continuation and what is the approximation error so to speak, by approximating with continuous model?). Did I miss anything? - Indeed I am surprised the authors do not cite works in control / system identification at all, where such linear learning systems have been studied (with much related techniques) for decades. Now, what makes it different from standard control of control-affine system is the fact that the control input is a matrix rather than a vector. However, is that not a case control / system ID has studied too? In summary, I am a bit suspicious of the relevance and novelty of this paper. However, as I am unfamiliar with the related literature in depth, I would say 'in dubio pro reo' with low confidence in my 'verdict'. Minor comments: -line 37 : ' minima' - The authors: explain the term 'continuation'.

Reviewer 4



The authors provide analysis of linear neural networks with and without regularization using the machinery of Hamilton's equations, and supplement their analysis with numerical results. Their work complements the growing body of literature on the behavior of the critical points of neural networks. Pros: The paper, while technically dense, is fairly easy to read, and the conclusions are clear--especially that of regularization drastically altering the landscape of critical points for linear(!) models. Cons: As always for papers concerning linear models, this reviewer would like some discussion of how these methods might be able to be used to tackle nonlinear networks (if at all). This reviewer also found Figure 1 quite difficult to parse, and would appreciate some extra language/figure engineering to make it clearer (a longer figure caption would probably be fine here). Finally, the authors appear to be analyzing solutions with regularization by first solving for the \lambda = 0 solution, and then locally extending this solution to study critical points near'' it with \lambda > 0. Why should one expect critical points near'' the \lambda = 0 solution to be the sorts of critical points found by a gradient descent algorithm? Said another way, this method seems to bias the analysis towards a certain subclass of critical points, but perhaps the reviewer is just confused. The reviewer would gladly increase the score upon further improvement and polishing of the manuscript.